# The Roles of the Virome in Cancer

**DOI:** 10.3390/microorganisms9122538

**Published:** 2021-12-08

**Authors:** Felix Broecker, Karin Moelling

**Affiliations:** 1Idorsia Pharmaceuticals Ltd., Hegenheimermattweg 91, CH-4123 Allschwil, Switzerland; 2Institute of Medical Microbiology, University of Zurich, Gloriastr. 30, CH-8006 Zurich, Switzerland; 3Max Planck Institute for Molecular Genetics, Ihnestr. 63-73, 14195 Berlin, Germany

**Keywords:** virome, microbiota, cancer, bacteriophages, fecal microbiota transplantation, checkpoint inhibitors, immunotherapy

## Abstract

Viral infections as well as changes in the composition of the intestinal microbiota and virome have been linked to cancer. Moreover, the success of cancer immunotherapy with checkpoint inhibitors has been correlated with the intestinal microbial composition of patients. The transfer of feces—which contain mainly bacteria and their viruses (phages)—from immunotherapy responders to non-responders, known as fecal microbiota transplantation (FMT), has been shown to be able to convert some non-responders to responders. Since phages may also increase the response to immunotherapy, for example by inducing T cells cross-reacting with cancer antigens, modulating phage populations may provide a new avenue to improve immunotherapy responsiveness. In this review, we summarize the current knowledge on the human virome and its links to cancer, and discuss the potential utility of bacteriophages in increasing the responder rate for cancer immunotherapy.

## 1. Introduction—The Human Virome

The human body hosts a large number of viruses, an estimated 10^12^–10^14^ particles per individual, that collectively are termed the virome [1,2]. Despite recent progress in determining the virome, a large fraction of viral sequence data collected in virome analysis studies remain unidentified ‘dark matter’. This indicates that a large number of currently unknown viruses, and their contributions to health and disease, remain to be characterized. Broadly speaking, the human virome consists of eukaryotic viruses, including pathogens that infect human cells, and viruses that infect bacteria termed bacteriophages or phages. Both human pathogenic viruses and phages have been linked to cancer, as will be discussed in this review. Other viruses that are also present in the virome, such as plant viruses (likely coming from food sources) will not be discussed here since no data are available on a possible connection to cancer. The by far largest component of the human virome are the phages, and most of them are associated with the bacteria they infect, which are most numerous in the intestinal tract, which harbors about 3.8 × 10^13^ bacteria per individual [3]. Here we summarize the current knowledge on the connection of the human virome to cancer development and discuss the possibility of modulating virus and phage populations to potentially increase the responder rates to cancer immunotherapy.

## 2. The Roles of the Eukaryotic Virome in Cancer

Various eukaryotic viruses infectious to humans have been implicated in carcinogenesis inside and outside the intestinal tract, mainly those of the families *Papillomaviridae* and *Herpesviridae* as well as hepatitis viruses (Table 1). In 2012, 15.4% of new cancers (2.2 million cases) worldwide were attributed to carcinogenic infections [4], most frequently with the bacterium *Helicobacter pylori* (770,000 cases) followed by 640,000 cases linked to human papillomavirus (HPV), 420,000 and 170,000 cases linked to hepatitis B and C virus (HBV and HCV), respectively, and 120,000 associated with Epstein–Barr virus (EBV, also known as human herpesvirus 4, HHV4). Of note, infections with these viruses does not necessarily lead to cancer. Rather, they often are one of many contributing factors to carcinogenesis. As such, virus-associated cancers typically develop as part of persistent infections over many years [5]. The proposed mechanisms of viral contributions to carcinogenesis also vary widely between viral species and will be briefly discussed in the following subsections. In general, viral infections can contribute to carcinogenesis by any of the following mechanisms: insertional mutation in the host genome, induction of inflammation and modulation of signaling pathways in the infected cells, for example, by viral oncogenes [6,7] (Figure 1).

### 2.1. Papillomaviridae

*Papillomaviridae* is a family of non-enveloped viruses with double-stranded DNA genomes [67]. Members infectious to humans, HPV, have been linked to various cancers, most significantly cervical cancer [11,12,13]. However, only a subset of the ~150 types of HPV, the so-called high-risk types such as HPV-16 and-18, are typically linked to cancer [68]. The main proposed mechanism of contribution of high-risk HPV to cervical cancer is through integration of the viral DNA into the host genome and expression of viral oncogenes. It has been shown, for example, that the E6 and E7 early oncogenes degrade tumor suppressors p53 and retinoblastoma protein (pRb), which among other effects cause the cell to arrest in the S-phase, leading to genomic instability, aneuploidy, DNA damage and, consequently, carcinogenesis [69,70] (see Figure 2 for details). It is thought that a similar mechanism is also at work in other HPV-associated cancers such as colorectal cancer [71]. HPV is the most frequently sexually transmitted infectious agent in the United States, and vaccines against HPV are recommended for all girls aged 11 to 12 years, aiming to reduce the burden of cervical cancer [72]. HPV has also been identified as a potential risk factor for various other cancer types, including anal [9,10], bladder [11] colorectal [14,15] esophageal [17,18,19], head and neck [11,20,21], oral [23], prostate [24], renal [25], skin/mucosal [26,27,28] and vulvar cancer [29]. In most of these cases, however, the association with HPV is not as strong as for cervical cancer, or is only seen in subsets of patients, and consequently there is also conflicting data. For example, a study from 2014 found no link between HPV infection and colorectal cancer [16]. In addition, there is an established correlation between HPV infection and head and neck cancer [11,20,21], but also data indicating that the presence of HPV in head and neck tumors predicts better long-term clinical outcome [22]. Moreover, merely associating the presence of specific viral sequences with cancer status does not necessarily indicate a causal role of the virus.

### 2.2. Herpesviridae

Viruses of the *Herpesviridae* family are enveloped dsDNA viruses infecting mammals, birds and reptiles [73]. Various members have been implicated in cancer formation, including cytomegalovirus (CMV, also known as human herpesvirus 5, HHV5), Epstein–Barr virus (EBV or HHV4), herpes simplex virus (HSV or HHV1/2) and HHV6 and HHV7 (Table 1). The strongest association with cancer has been reported for EBV, which has been linked to colorectal carcinoma (CRC) [33] (although a later study found no association [16]), esophageal [34] and gastric cancer [35] (interestingly, however, patients with EBV-positive gastric cancer had a better response to chemotherapy and better survival [36]), hepatocellular carcinoma [37], lymphoma (Burkitt lymphoma, diffuse large B-cell lymphoma and peripheral T-cell lymphoma) [38,39,40], oral [41], as well as skin and mucosa associated cancers [26]. Therefore, it has been proposed that vaccination against EBV might be a viable means to prevent EBV-associated cancers [74]. Various EBV vaccine candidates are currently in preclinical or clinical development.

CMV has been linked to colorectal cancer [30], however, in non-elderly patients CMV-positive tumors have been associated with increased disease-free survival rate [31]). HHV6 was found to be connected to malignant melanoma [26] and B-cell lymphoma [39], HHV7 to bladder and oral cancer as well as to T-cell lymphoma [26], and HSV has been associated with oral cancer [41].

In HIV-infected individuals, Kaposi sarcoma-associated herpesvirus (KSHV) infection is associated with Kaposi sarcoma [42].

Mechanistically, herpesviruses such as CMV and HSV and others are known to induce DNA synthesis and counteract apoptosis via activation of the rat sarcoma (Ras)/rat fibrosarcoma (Raf)/meiosis-specific kinase (MEK)/extracellular signal-related kinase (ERK) pathway [71,75] (see Figure 3 for details). In addition, KSHV has been shown to inhibit tumor suppressor protein p53 [76].

### 2.3. Polyomaviridae

Two members of the *Polyomaviridae* family (non-enveloped double-stranded DNA (dsDNA) viruses), namely BKV and JCV, have been linked to CRC [43,44,46,47,48,49]. However, other studies have challenged this notion and found no link [16,45]. There is also a possible association of BKV with bladder cancer [26]. The clearest evidence, however, has been obtained for Merkel cell polyomavirus (MCV) that has been identified as a major causative factor for Merkel cell carcinoma [50,51]. Consequently, it has been proposed that an anti-MCV vaccine might help to prevent Merkel cell carcinoma [77,78].

It has been hypothesized that MCV contributes to oncogenesis in two steps: integration of the viral genome into the host genome followed by mutational truncation of the large T antigen gene [79]. The truncated T antigen promotes cell cycle progression. The T antigen likely also plays a role in oncogenic transformation mediated by other polyomaviruses such as BKV and JCV. It has been shown to interact with various host proteins, thereby altering the processes of DNA repair, cell cycle regulation and proliferation [80] (see Figure 4 for details).

### 2.4. Retroviridae

The retroviruses human immunodeficiency virus (HIV) and human T-lymphotropic virus type 1 (HTLV-1) have both been associated with cancer. Three HIV-associated cancers are known, cervical cancer, aggressive B cell non-Hodgkin lymphoma and Kaposi sarcoma [55,56]. In addition, HIV infection has been linked to an increased risk for anal cancer [52,53] and worse overall colostomy-free survival rates [54]. The effect of HIV infection is likely indirect; the virus weakens the immune system and renders infected individuals more susceptible to infections by other viruses, such as Kaposi sarcoma-associated herpesvirus (KSHV), which has been linked to Kaposi sarcoma [42]. For HTLV-1, oncogenesis is thought to involve multiple mechanisms [6,81]. These include: (1) the establishment of a persistent infection that results in chronic inflammation, which attracts phagocytes that release reactive oxygen species, contributing to DNA damage, (2) the genomic integration and expression of viral oncogenes, such as the Tax-1 protein that, among other functions, inhibits tumor suppressor p53, and (3) the suppression of host immune responses. A summary of the functions of Tax-1 is shown in Figure 5.

### 2.5. Other Viruses

Various other viruses have been linked to cancer. Among these are the hepatitis viruses HBV and HCV. HBV (family *Hepadnaviridae* with a partially double-stranded DNA genome and an envelope) infects the liver and has been linked to liver cancer [11] but may also be a risk factor for cancers of the bile duct [59], the colon [60] and the pancreas [61,62]. HCV (an enveloped ssRNA virus of the *Flaviviridae* family) also infects the liver and has been associated with liver and bile duct cancer [11,59]. Liver carcinogenesis by HBV and HCV is multifactorial and has been linked, amongst others, to virus-induced changes in signaling pathways, the inflammatory status and to elevated production of reactive oxygen species (ROS) and reactive nitrogen species (RNS) [82,83,84] (see Figure 6 for details). Anelloviruses (circular single-stranded DNA (ssDNA) viruses) including torque teno virus (TTV) have been associated with hepatic [63] and mucosal [26] cancer as well as leukemias [26]. Human bocavirus (HBoV) and other members of the *Parvoviridae* family (ssDNA viruses) have been linked to various cancers, including colorectal and lung cancer [64], squamous cell carcinoma [65] and skin cancer [26]. The ssRNA orthobunyaviruses may be linked to colorectal cancer [66]. The mechanisms of oncogenesis of anelloviruses, parvoviruses and orthobunyaviruses remain largely unknown.

## 3. The Human Intestinal Virome and Its Links to Cancer

The human intestinal microbiota is a complex community of bacteria, archaea, viruses, fungi, and eukaryotic parasites [85]. Collectively, the microbes outnumber human cells. The human body contains about 3 × 10^13^ human cells and 3.8 × 10^13^ bacteria [3], and bacterial genes (~2,000,000) are estimated to be about 100 times more numerous than human genes (~20,000) [86]. Viruses, predominantly phages, are likely the most abundant biological entities, with an estimated number of 10^15^ in the human gastrointestinal tract [87]. Other community members such as archaea of fungi are substantially less abundant. Collectively, one could take the view that we are symbionts of host and microbial cells, each with their own set of genes. The gut microbiota plays important roles in food digestion, development of the immune system and protection against microbial pathogens. Pathological changes in the microbial composition (dysbiosis), induced by infections, chronic diseases, or antibiotic treatment, can lead to failure of control of pathogens and severe damage as in inflammatory bowel disease (IBD). Dysbiosis is also observed in extra-intestinal diseases such as asthma or autism [88].

The composition of the human virome is largely unknown. A recent study described a catalog of tens of thousands of viruses from human metagenomes, which revealed associations with chronic diseases [89]. This study describes 45,000 unique viruses as part of the human microbiota, and about 2000 specific phages were found to correlate with a variety of common chronic diseases such as Parkinson’s disease and obesity. All bacteria are infected by phages, which play a role in maintaining the bacterial population in healthy individuals. Depending on environmental conditions and the genetic composition of the phages, they can lyse their bacterial hosts. In the oceans where bacteria and phages are very abundant lysis occurs in about 20% of the bacteria in about 24 h [90]. The virome may regulate the microbiome and may influence bacterial complexity, whereby the population dynamics likely follow a ‘kill-the-winner’ ecological model. In this model, the most abundant bacteria are killed by their phages, other bacterial taxa will take over the ecological niche and be subsequently killed by their phages [90]. Thereby, phages play a crucial role in shaping the composition of the intestinal bacterial communities. They also facilitate horizontal gene transfer and thereby the functional capacity of the microbiota [90]. The virome is more challenging to study and to characterize than bacterial populations, and many of its contributions to ecosystems are still poorly understood. Many of the viruses are unknown, sometimes designated as ‘viral dark matter’ [91].

Changes in the fecal virome, especially the phage population, have been reported recently for CRC patients [7,66,92,93]. It has been suggested that phages may play a causative role in CRC by altering the bacterial populations of the intestine such that pathogenic bacteria can thrive and form biofilms [92,94] (Figure 7). Several phage species have been linked to CRC [66,71]. Comparing the fecal microbiota of CRC patients to healthy controls, specific phages have been found to be more abundant in patients that could play a role in shifting the overall bacterial populations or serve as biomarkers. Phage SpSL1, for example, infects bacteria of the *Streptococcus* genus and was found to be more abundant in early-stage CRC than in controls but its abundance decreased in later stages [66]. Depletion of commensal *Escherichia coli* bacteria by various phages whose abundance increases during CRC, including *Enterobacteria* phage HK544, Punalikevirus, Lambdalikevirus and Mulikevirus, may contribute to the disease-associated dysbiosis [66]. However, it remains unknown if phages have a causal role in the development of dysbiosis, or whether their increased abundance is a consequence of dysbiosis. The observed differences, however, might serve as a starting point for targeted microbiota manipulations aiming to reverse the dysbiosis, which may have a positive impact on treatment.

## 4. Fecal Microbiota Transplantation—Focus on Viruses and Cancer

In recent years, fecal microbiota transplantation (FMT), has gained attention mostly as a remedy for recurrent *Clostridioides difficile* infections, with impressive cure rates of about 90% or higher [96]. We described such a case of a patient from Zurich, Switzerland who received fecal material from a healthy donor and recovered without adverse events, meanwhile healthy for almost ten years [97]. An indicator of successful therapy is the diversity of the microbiota, which is higher and different in composition in healthy people than in *C. difficile* patients where it is diminished due to antibiotic therapy.

Recent data suggest that sterile fecal filtrate in which bacteria have been removed (but phages retained) can also effectively cure *C. difficile* infections [98,99]. Moreover, successful FMT was associated with the stable transfer of phages from donor to recipient [100,101,102]. We have shown that a core virome comprising the most dominant phage species is transmitted during FMT and stabilizes in the recipient before the bacterial populations do [101]. These studies indicate that phages might be sufficient in conferring the beneficial effects of FMT, although a contribution of metabolites, proteins and bacterial fragments present in the filtrate cannot be excluded.

The composition of the intestinal microbiota has recently been shown to contribute to the success of anticancer therapy with checkpoint inhibitors [103]. The new immunotherapy has a 20–40% response rate, but in the remaining patients the therapy has virtually no effect [104]. How to cure the remaining 60–80% is one of the most urgent research topics. The microbiota likely plays a role. It has been shown recently that FMT using donor stool from responders can boost the response to checkpoint inhibitor therapy of previous non-responder melanoma patients [105,106]. Conversely, antibiotic therapy, which disrupts the intestinal microbiota, correlates with poor outcome of checkpoint inhibitor therapy [107].

Germ-free or antibiotic-treated mice with depleted intestinal bacteria serve as animal models for the effects of the microbiota of responders and non-responders [103]. Indeed, instilling microbiota from responders transferred a “responder” phenotype (i.e., relatively slow tumor growth) to mice, whereas transfer from non-responders rendered mice non-responsive (i.e., relatively fast tumor growth) [108,109,110,111]. The comparison of stool samples from responders and non-responders revealed certain bacterial species whose presence or abundance correlated with successful treatment. This included, among others, *Faecalibacterium prausnitzii*, *Akkermansia muciniphila*, *Bifidobacterium longum* and *Bacteroides caccae*, depending on the type of cancer [108,109,110,111,112].

Stool transfer from a responder can help to convert a non-responder into a responder. This was shown recently by two studies [105,106]. Davar et al. [105] performed FMT on 15 non-responding patients with malignant melanoma using donor stool from responders. Six of the 15 patients showed objective responses to pembrolizumab, a monoclonal antibody against programmed cell death-1 (PD-1), following the procedure. There were also no severe side effects. Bacteria associated with clinical response were, among others, *Bifidobacterium longum*, *Colinsella aerofaciens* and *Faecalibacterium prausnitzii*, enriched were mostly bacteria of the *Firmicutes* phylum (*Ruminococcaceae* and *Lachnospiraceae* families). In addition, response to therapy correlated with a specific subset of T cells with cytolytic function (CD56^+^CD8^+^ T cells) and decreased interleukin-8 expressing myeloid cells. Some of the above-mentioned bacteria were confirming earlier observations [105]. Thus, a single FMT helped to overcome primary resistance to immunotherapy in a subset of melanoma patients. Why still nine out of 15 patients remained non-responsive remains unclear.

In a similar study, ten patients with anti-PD-1 refractory metastatic melanoma received FMT with donor stool from responders followed by reintroduction of anti-PD-1 immunotherapy [106]. Two partial responses and one complete remission were observed. Patients first underwent a microbiota depletion phase with antibiotic therapy for 72 h (vancomycin, neomycin). They received oral stool capsules every two weeks for 90 days and anti-PD-1 therapy with nivolumab. No clear-cut microbiota indicative for responders was determined. The safety of FMT was confirmed. Antibiotic therapy was effective as pre-FMT protocol and was intended to allow for a better engraftment of the transferred microbiota.

Of note, the viromes of responders and non-responders so far have not been analyzed, so it remains unclear if there are certain phage species or eukaryotic viruses that correlate with successful treatment. However, it has been shown recently that a phage-specific T cell epitope is linked to PD-1 responsiveness in mice [113]. This T cell epitope has been proposed to induce memory T cells cross-reacting with a tumor antigen. This opens up the avenue that certain phage species could be used therapeutically to stimulate the immune system to enhance the efficacy of immunotherapy. Moreover, the microbiota of responders may contain such phage species which activate T cells that are cross-reactive to tumor antigens; phages that may be absent in non-responders. Modulating phage populations of non-responders may therefore be a promising avenue to increase responsiveness to immunotherapy. Eukaryotic viruses may also contribute to responsiveness through their known roles in modulating the immune system [114]. Virome analyses of responders and non-responders are warranted to identify such phages and viruses.

## 5. Conclusions and Outlook

Carcinogenesis is a multifactorial process that involves genetic and environmental factors, the microbiota, and the virome [115] (Figure 8). Both eukaryotic viruses and phages can contribute to cancer formation and progression, as discussed herein, through various mechanisms. Of note, we are only beginning to understand the complexity of the human virome, and new viruses will be discovered in the ‘dark matter’ of so far unknown virus-like sequences generated during virome analyses. These may include viruses which contribute to cancer that we may, for instance, ingest with our food [116], or the more recently discovered giant viruses including *Mimiviridae* and *Phycodnaviridae* whose oncogenic potential may be underestimated [117,118]. Knowing the identity of potential oncogenic viruses would provide the opportunity for intervention and vaccination approaches that have already been implemented for HPV to effectively prevent cervical cancer. The virome may also play a role in the response of patients to cancer immunotherapy with checkpoint inhibitors; similar to beneficial bacterial taxa that correlate with response to treatment, beneficial viruses and phages may be identified. In particular, phages may be crucial in mediating the effects of FMT, a therapeutic intervention that has shown promise to ‘convert’ non-responders to immune therapy to responders [105,106]. We and others have shown that during FMT, phages are transmitted [100,101,102,119]. In some indications, sterile fecal filtrates (containing phages, but not bacteria) may even be more efficient than FMT, as shown recently for necrotizing enterocolitis of preterm infants [120]. The role of phages and eukaryotic viruses in determining the responder status to immunotherapy has not been investigated to date. This knowledge may help to improve response rates for cancer immunotherapy by targeted supplementation and modulation of the patients’ phageomes and viromes.

## Figures and Tables

**Figure 1 microorganisms-09-02538-f001:**
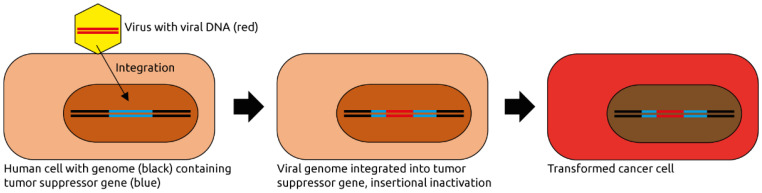
Eukaryotic virus-induced carcinogenesis. A virus infects a human cell and integrates its genome into the host cell genome. In this example, the viral genome integrates into and thereby inactivates a tumor suppressor gene, contributing to oncogenic transformation. The viral genome may also express viral oncogenes, or it activates a near-by host proto-oncogene (not shown). Figure adapted from Marônek et al. [7].

**Figure 2 microorganisms-09-02538-f002:**
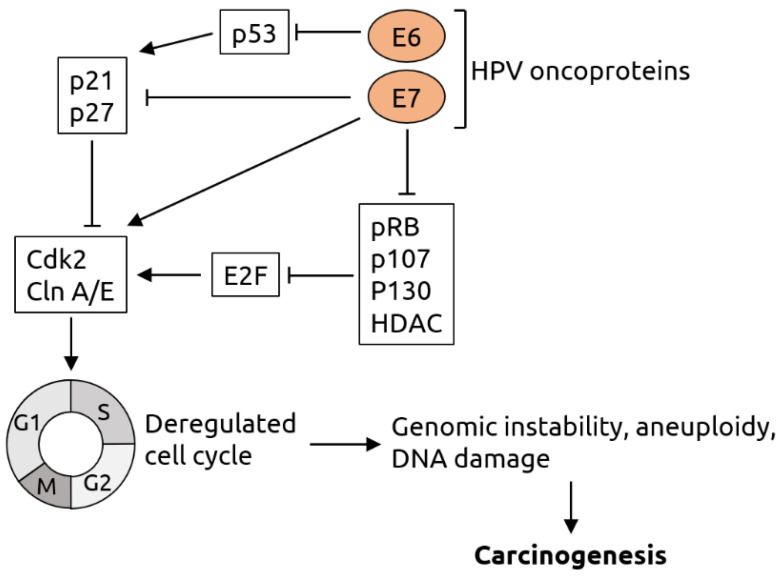
Molecular mechanisms of HPV-induced carcinogenesis. The viral oncoprotein E6 inhibits tumor suppressor p53 and thereby indirectly suppresses the p21 and p27 proteins, which are negative regulators of cyclin-dependent kinase 2 (Cdk2) and cyclin (Cln) A/E. Cdk2 and Cln A/E are important factors for cell cycle regulation. The viral oncoprotein E7 inhibits p21 and p27 and directly activates Cdk2 and Cln A/E. In addition, E7 inhibits retinoblastoma protein pRB and related pocket proteins p107 and p130 as well as specific histone deacetylases (HDAC). pRB, p107, p130 and HDAC are inhibitors of E2F transcription factors, which activate Cdk2 and Cln A/E. Thus, E6 and E7 act synergistically in cell cycle deregulation, which results in genomic instability, aneuploidy and DNA damage and, consequently, carcinogenesis. Figure adapted from Lehoux et al. [70].

**Figure 3 microorganisms-09-02538-f003:**
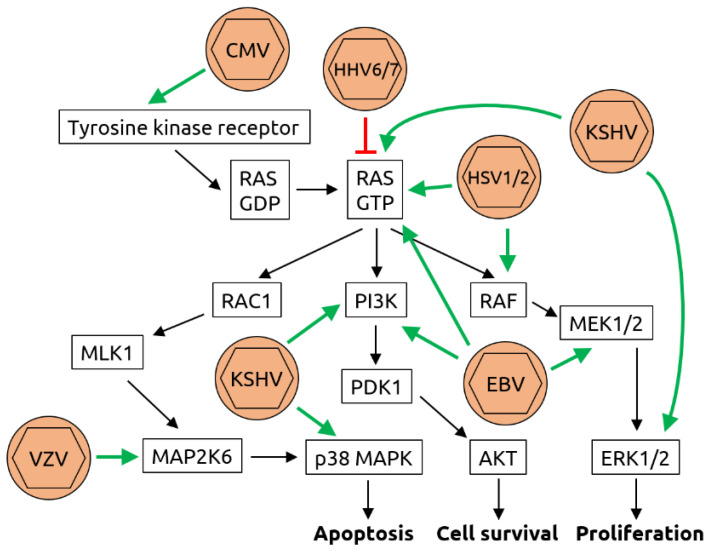
Molecular mechanisms of herpesvirus-induced carcinogenesis. Shown is a graphical representation of the key mediators of the Ras/Raf/MEK/ERK pathway. The known interactions between herpesviruses and proteins of this pathway are indicated by green arrows (activation) or red T-bars (inhibition). Cytomegalovirus (CMV) has been shown to bind to and activate tyrosine kinase receptors such as epidermal growth factor receptor (EGFR). While human herpesviruses (HHV) 6 and 7 inhibit rat sarcoma (RAS) bound to guanosine triphosphate (RAS-GTP), herpes simplex viruses 1 and 2 (HSV1/2), Kaposi sarcoma-associated herpesvirus (KSHV) and Epstein–Barr virus (EBV) all activate RAS-GTP. In addition, KSHV activates extracellular signal-regulated kinases 1/2 (ERK1/2), phosphoinositide 3-kinase (PI3K) and p38 mitogen-activated protein kinase (MAPK). EBV also activates PI3K as well as meiosis-specific kinases MEK1/2 and HSV1/2 activate rat fibrosarcoma (RAF). Although an association of varicella zoster virus (VZV, also known as HHV3) with cancer is not established, it has been shown to be capable of transforming cells in vitro and to activate dual specificity mitogen-activated protein kinase 6 (MAP2K6). Through these manipulations of the pathway, the processes of apoptosis, cell survival and proliferation are deregulated, contributing to carcinogenesis. Figure adapted from Filippakis et al. [75], a review in which details can be found.

**Figure 4 microorganisms-09-02538-f004:**
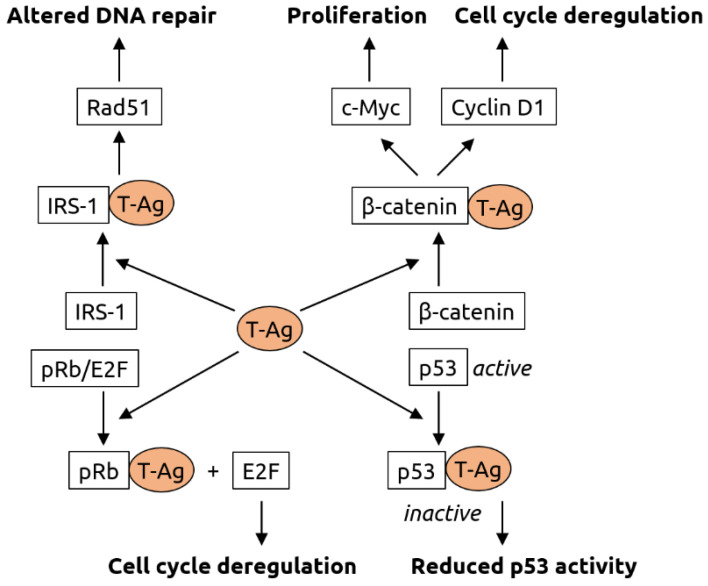
Molecular mechanisms of carcinogenesis induced by the T-antigen (T-Ag) of polyomaviruses. T-Ag has been shown to bind to insulin receptor substrate 1 (IRS-1), causing its translocation into the nucleus, where it is likely involved Rad51 trafficking. Rad51 is known to be involved in DNA repair. T-Ag also associates with β-catenin, leading to its translocation into the nucleus. There, β-catenin activates the c-Myc proto-oncogene (involved in cell proliferation) and cyclin D1 (involved in cell cycle regulation). Interaction of T-Ag with retinoblastoma protein pRb leads to the release of transcription factor E2F that is involved in cell cycle regulation. T-Ag also inhibits tumor suppressor p53. Figure adapted from White and Khalili [80], a review in which details can be found.

**Figure 5 microorganisms-09-02538-f005:**
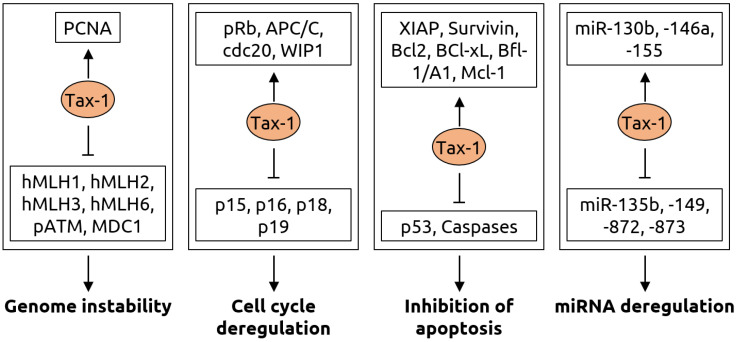
Molecular mechanisms of carcinogenesis induced by the Tax-1 protein of HTLV-1. Genome instability is caused by Tax-1 through activation of proliferating cell nuclear antigen (PCNA) and suppression of hMLH mismatch repair proteins, ataxia-telangiectasia mutated (ATM) phosphorylation (pATM) and mediator of DNA damage checkpoint 1 (MDC1). Cell cycle deregulation occurs via activation of retinoblastoma protein pRb, anaphase-promoting complex/cyclosome (APC/C), its binding partner cdc20 and wild-type p53-induced phosphatase 1 (WIP1) as well as inhibition of p15, p16, p18 and p19, which are inhibitors of cyclin-dependent kinase 4 (CDK4). Tax-1 leads to evasion of apoptosis by activating various anti-apoptotic proteins including X-linked inhibitor of apoptosis (XIAP), Survivin, the B-cell lymphoma (Bcl) family proteins Bcl-2 and Bcl-xL, Bcl-2-related protein Bfl-1/A and myeloid cell factor-1 (Mcl-1) as well as by suppressing tumor suppressor p53 and Caspase-3, -7, -8 and-9. Furthermore, Tax-1 alters numerous microRNAs resulting in a fine-tuning of gene expression for oncogenic transformation, with an upregulation of miR-130b, -146a, -155 and downregulation of miR-135b, -149, -872 and-873. Figure adapted from Mohanty and Harjah [81], a review in which details can be found.

**Figure 6 microorganisms-09-02538-f006:**
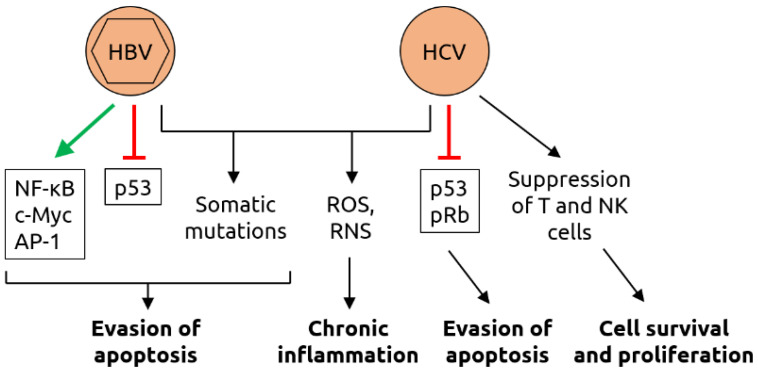
Molecular mechanisms of carcinogenesis induced by HBV and HCV. HBV infection leads to the activation of various transcription factors, including nuclear factor of activated B cells (NF-κB), c-Myc and activator protein 1 (AP-1), and the suppression of tumor suppressor p53 which, together with virus-induced somatic mutations, contributing to evasion of apoptosis. As HBV, HCV infection induces somatic mutations and increases the levels of reactive oxygen species (ROS) and reactive nitrogen species (RNS); ROS and RNS contribute to chronic inflammation. HCV also suppresses p53 and retinoblastoma protein pRb, contributing to evasion of apoptosis. Suppression of T cells and natural killer (NK) cells by HCV support cell survival and proliferation. Figure adapted from Karpiński [84], a review in which details can be found.

**Figure 7 microorganisms-09-02538-f007:**
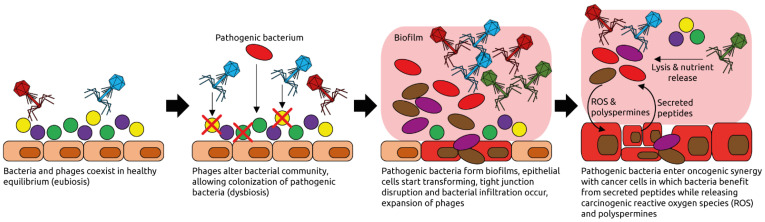
Examples of phage-induced carcinogenesis. Phages coexist with their host bacteria in a well-balanced equilibrium, for example in the intestinal tract. Some phages might get activated pathologically and alter the bacterial community structure through lysis. Pathogenic bacteria can then thrive and form biofilms. Phages populations expand, typical for dysbiotic, intestinal microbiota and inflammation [95]. Phages lyse commensal bacteria, releasing nutrients that are used by the pathogenic ones. Reactive oxygen species (ROS) and polyspermines produced in the biofilms contribute to DNA damage of host cells, and thereby oncogenic transformation [94]. Peptides released by the transformed cells are metabolized by the pathogenic bacteria. Figure adapted from Hannigan et al. [92].

**Figure 8 microorganisms-09-02538-f008:**
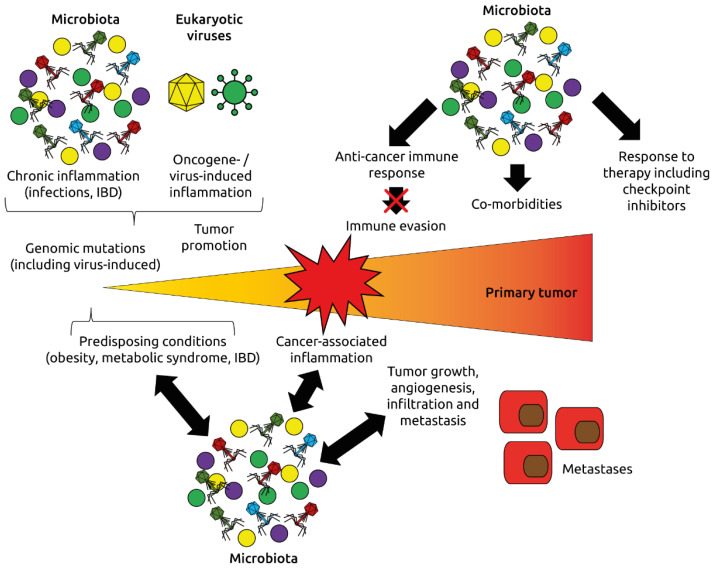
Roles of the microbiota and virome (eukaryotic viruses and phages) in oncogenesis and response to immunotherapy. The microbiota and virome affect inflammation, one of the hallmarks of cancer, the development of cancer-promoting conditions such as obesity, metabolic syndrome and inflammatory bowel disease (IBD), and modulate immune mechanisms regulating cancer initiation and progression. Adapted from Dzutsev et al. [115].

**Table 1 microorganisms-09-02538-t001:** Eukaryotic viruses linked to cancer. Abbreviations: ATLL, adult T-cell leukemia/lymphoma; BKV; BK polyomavirus; CMV, cytomegalovirus; CTCL, cutaneous T-cell lymphoma; DLBCL, diffuse large B-cell lymphoma; EBV, Epstein–Barr virus; HBoV; human bocavirus; HBV, hepatitis B virus; HCC, hepatocellular carcinoma; HCV, hepatitis C virus; HHV, human herpesvirus; HIV, human immunodeficiency virus; HPV, human papillomavirus; HSV, herpes simplex virus; HTLV-1, human T-lymphotropic virus type 1; JCV, JC polyomavirus; KSHV, Kaposi sarcoma-associated herpesvirus; MCV, Merkel cell polyomavirus; NHL, non-Hodgkin lymphoma; PTCL, peripheral T-cell lymphoma; SCC, squamous cell carcinoma; TTV, torque teno virus.

Virus Family	Virus	Cancer Type	Observations
*Papillomaviridae*	HPV-16	Anal	HPV, especially HPV-16, is a possible risk factor for anal and rectal cancer [8,9] and a significant prognostic marker, especially for locally advanced disease [10]
	HPV	Bladder	HPV (different serotypes) may be linked to bladder cancer in a small number of cases [11]
	HPV-16, -18	Cervical	Association between infection with high-risk HPV serotype (mainly HPV-16 and-18) and development of cervical cancer [11,12,13]
	HPV-18	Colorectal	HPV, especially HPV-18, is a possible risk factor for colorectal cancer [14,15], however, another study found no association [16]
	HPV-16, -18, -26, -57	Esophageal	HPV-16 is a risk factor for esophageal carcinoma [17,18]; HPV infection (mainly HPV-16, -18, -26 and-57) is common in esophageal carcinoma [19]
	HPV-16	Head and neck (SCC)	HPV infection, especially HPV-16, is associated with head and neck cancer [11,20,21] and better long-term outcome [22]
	HPV-6	Oral	Association of HPV-6 with oral cancer [23]
	HPV-16	Prostate	Association of HPV-16 with prostate cancer [24]
	HPV-16, -18, -58	Renal	Association of HPV-16, -18 and-58 with renal cell carcinoma [25]
	HPV-5, -8	Skin and mucosal	Papillomavirus DNA frequently detected in skin-and mucosa-associated cancers [26]; HPV-5 and-8 are associated with epidermodysplasia verruciformis associated with a high risk of skin cancer [27,28]
	HPV-16	Vulvar	Association between HPV, especially HPV-16, and vulvar squamous cell carcinoma [29]
*Herpesviridae*	CMV (HHV5)	Colorectal	CMV DNA is more abundant cancer tissues compared to healthy tissues [30]; CMV-positive tumors in non-elderly patients are associated with increased disease-free survival rate [31]; specific genetic polymorphisms of CMV are linked to different clinical outcomes [32]
	EBV (HHV4)	Colorectal	Possible association of EBV with colorectal carcinoma [33], however, no association found in another study [16]
	EBV (HHV4)	Esophageal	EBV is associated with esophageal cancer [34]
	EBV (HHV4)	Gastric	Possible involvement of EBV in gastric cancer and precursor lesions [35]; patients with EBV-positive gastric cancer had a better response to chemotherapy and better survival [36]
	EBV (HHV4)	Hepatic	EBV infections detected in HCC tissues [37]
	EBV (HHV4)	Lymphoma (Burkitt)	EBV infections contribute to Burkitt lymphoma [38]
	EBV (HHV4)	Lymphoma (DLBCL)	EBV RNA detected in B-cell lymphoma samples [39]
	EBV (HHV4)	Lymphoma (PTCL)	EBV expression associated with some subtypes of peripheral T-cell lymphomas [40]
	EBV (HHV4)	Oral	Higher proportion of EBV-positive oral squamous cell carcinoma in industrialized countries [41]
	EBV (HHV4)	Skin and mucosal	EBV DNA frequently detected in skin and mucosal cancers [26]
	HHV6	Lymphoma (DLBCL)	HHV6 RNA detected in B-cell lymphoma samples [39]
	HHV6	Malignant melanoma	HHV6 DNA frequently detected in malignant melanoma [26]
	HHV7	Bladder	HHV7 DNA frequently detected in bladder cancer [26]
	HHV7	Lymphoma (CTCL)	HHV7 DNA frequently detected in cutaneous T-cell lymphoma (Mycosis fungoides) [26]
	HHV7	Oral	HHV7 DNA frequently detected in oral cavity cancer [26]
	HSV (HHV1/2)	Oral	Higher proportion of HSV-positive oral squamous cell carcinoma in industrialized countries [41]
	KSHV (HHV8)	Kaposi sarcoma	In HIV-infected individuals, KSHV infection is associated with Kaposi sarcoma [42]
*Polyomaviridae*	BKV	Bladder	Possible association of BKV with bladder cancer [26]
	BKV	Colorectal	Possible association of BKV with colorectal cancer [43,44], however, other studies found no association [16,45]
	JCV	Colorectal	JCV is associated with colorectal cancer [45,46] and may be involved in carcinogenesis [47], specifically in chromosomal instability [48]; JCV T-antigen is expressed in early-stage colorectal cancer [49], however, another study found no association [16]
	MCV	Merkel cell carcinoma	MCV is the major causative factor for Merkel cell carcinoma [50,51]
*Retroviridae*	HIV	Anal	HIV-positive people have increased risk for anal cancer [52,53] and worse overall colostomy-free survival rates [54]
	HIV	Cervical	Cervical cancer is more prevalent in HIV-positive individuals, likely because of increased susceptibility to HPV infection [55,56]
	HIV	Kaposi sarcoma	Kaposi sarcoma is more prevalent in HIV-positive individuals, likely because of increased susceptibility to KSHV infection [55,56]
	HIV	Lymphoma (NHL)	Aggressive B cell non-Hodgkin lymphoma is more prevalent in HIV-positive individuals, likely because of increased susceptibility to EBV infection [55,56]
	HTLV-1	Lymphoma (ATLL)	HTLV-1 induces adult T-cell leukemia/lymphoma in 5% of infected individuals [57] through random integration into the host genome [58]
Others	HBV	Bile duct	HBV is a risk factor for bile duct cancer [59]
	HBV	Colorectal	Chronic HBV infection is a risk factor for colorectal cancer [60]
	HBV	Hepatic	Liver cancer is associated with HBV [11]
	HBV	Pancreatic	Chronic HBV infection [61] or past exposure [62] are risk factors for pancreatic cancer
	HCV	Bile duct	HCV is a risk factor for bile duct cancer [59]
	HCV	Hepatic	Liver cancer is associated with HCV [11]
	TTV	Hepatic	TTV is a risk factor for hepatocellular carcinoma [63]
	HBoV	Colorectal	Some colorectal cancers are associated with HBoV [64]
	HBoV	Lung	Some lung cancers are associated with HBoV [64]
	HBoV	Tonsillar	Association of HBoV with tonsil squamous cell carcinoma [65]
	Orthobunyaviruses	Colorectal	High abundance of orthobunyaviruses in colorectal cancer [66]
	Parvoviruses	Skin	Parvovirus DNA frequently detected in skin-associated cancers [26]
	Anelloviruses	Mucosal	Anellovirus DNA frequently detected in mucosal cancers [26]
	Anelloviruses	Leukemias	Anellovirus DNA frequently detected in leukemias [26]

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
