# Peer review of "The Roles of the Virome in Cancer"

_microorganisms, 2021, doi:10.3390/microorganisms9122538_

Round 1
Reviewer 1 Report
Congratulations for Authors. You wrote very good article, which I suggest publish after adding more detailed figure with cancerogenic mechanisms of action of described viruses. Please see and you can cite the following article https://www.sciencedirect.com/science/article/abs/pii/S0889855319300342?via%3Dihub, in which are more detailed picture for HBV and HCV, with increase or decrease of specific compounds or cell, giving specific effect.
Author Response
Congratulations for Authors. You wrote very good article, which I suggest publish after adding more detailed figure with cancerogenic mechanisms of action of described viruses. Please see and you can cite the following article https://www.sciencedirect.com/science/article/abs/pii/S0889855319300342?via%3Dihub, in which are more detailed picture for HBV and HCV, with increase or decrease of specific compounds or cell, giving specific effect.
Response: Thank you very much for your valuable comments! We have included additional figures on the carcinogenic mechanisms (Figures 2-6 for HPV, herpesviruses, polyomaviruses, HTLV-1 and HBV + HCV, respectively), as suggested. We now also cite the mentioned publication.
Reviewer 2 Report
In this review manuscript Broeker and Moelling summarize the carcinogenic aspect of the human virome, devoting paragraphs to both common human viral pathogens, which can potentially take on a role in oncogenesis, and viruses of the human microbiome, occupying the digestive tract and other organs. The review cites an impressive number of research articles and is written in clear, easy-to-comprehend English. Especially with regards to the broad scope of the journal the review was submitted to, the manuscript fits well into this and has merit. Some questions and issues raise, however, which should be addressed.
Even the manuscript itself decalares that our knowledge on how bacteriophages are associated with cancer is rather limited, yet half of the manuscript is devoted to this topic, much of it focusing on FMT. The paragraphs on the human oncogenic viruses, on which field there is an impressive deal of research published, feel rushed in contrast. There are several types of cancers and instances listed here these viruses are associated with, yet only a very limited description on the actual mechanism. The review would greatly benefit from discussing these in greater detail and more specifically to each virus type, not just vaguely for the whole family. Even the addition of a figure, summarizing the mechanisms of viral oncogenesis, would improve the manuscript significantly.
As this article is intended to be a review, extra attention needs to be paid to ensure that all abbreviations are resolved, giving an easier time for readers of other disciplines to comprehend it. There are several abbreviations throughout (including virus names, cancers, pathways, therapies), which remain unresolved.
Specific points:
Figure 1: Figure 1A and 1B should be separate figures, as fig. 1B is only referenced way later in the text
Table 1: The authors later state that only certain HPV types are of high risk, yet serotype numbers are only listed in case of herpesviruses. Members of the other families should be listed with the serotype numbers as well, especially given that some of these are actually mentioned in the text later on.
When talking about Epstein-Barr virus, there are no mentions of Burkitt lymphoma, yet two other lymphomas are described. This should also be added to the respective paragraph.
Line 130: The large T antigen is mentioned but not its actual role in oncogenesis. More details should be provided on this.
Line 150: Although the HBV genome does have a partial dsDNA segment to prime reverse transcription, this virus is not a dsDNA virus, but a member of the Riboviria realm along with all RNA viruses and a member of a separate Baltimore class.
Paragraph 4: Although the topic of FMT is certainly intriguing, the authors themselves state that the virome of bacteriphages associated with it has never been investigated. I suggest to decrease the length of this chapter and limit it to the studies where the viral involvement was actually confirmed (such as the sterile filtration aspect etc.)
Line 256: What is the responder/ non-responder phenotype?
Author Response
In this review manuscript Broeker and Moelling summarize the carcinogenic aspect of the human virome, devoting paragraphs to both common human viral pathogens, which can potentially take on a role in oncogenesis, and viruses of the human microbiome, occupying the digestive tract and other organs. The review cites an impressive number of research articles and is written in clear, easy-to-comprehend English. Especially with regards to the broad scope of the journal the review was submitted to, the manuscript fits well into this and has merit. Some questions and issues raise, however, which should be addressed.
Even the manuscript itself decalares that our knowledge on how bacteriophages are associated with cancer is rather limited, yet half of the manuscript is devoted to this topic, much of it focusing on FMT. The paragraphs on the human oncogenic viruses, on which field there is an impressive deal of research published, feel rushed in contrast. There are several types of cancers and instances listed here these viruses are associated with, yet only a very limited description on the actual mechanism. The review would greatly benefit from discussing these in greater detail and more specifically to each virus type, not just vaguely for the whole family. Even the addition of a figure, summarizing the mechanisms of viral oncogenesis, would improve the manuscript significantly.
Response: As suggested, we have included more detailed descriptions on the mechanisms of oncogenesis by adding five new figures (Figures 2-6 for HPV, herpesviruses, polyomaviruses, HTLV-1 and HBV + HCV, respectively). The text has been changed accordingly.
As this article is intended to be a review, extra attention needs to be paid to ensure that all abbreviations are resolved, giving an easier time for readers of other disciplines to comprehend it. There are several abbreviations throughout (including virus names, cancers, pathways, therapies), which remain unresolved.
Response: We carefully went through the manuscript and believe that now all abbreviations are explained.
Specific points:
Figure 1: Figure 1A and 1B should be separate figures, as fig. 1B is only referenced way later in the text
Response: As suggested, we split this figure into two (Figures 1 and 7).
Table 1: The authors later state that only certain HPV types are of high risk, yet serotype numbers are only listed in case of herpesviruses. Members of the other families should be listed with the serotype numbers as well, especially given that some of these are actually mentioned in the text later on.
Response: We added serotype information to Table 1.
When talking about Epstein-Barr virus, there are no mentions of Burkitt lymphoma, yet two other lymphomas are described. This should also be added to the respective paragraph.
Response: Burkitt lymphoma has been added to the table and text.
Line 130: The large T antigen is mentioned but not its actual role in oncogenesis. More details should be provided on this.
Response: Functions of the large T antigen are now described in detail in the new Figure 4.
Line 150: Although the HBV genome does have a partial dsDNA segment to prime reverse transcription, this virus is not a dsDNA virus, but a member of the Riboviria realm along with all RNA viruses and a member of a separate Baltimore class.
Response: We corrected this accordingly.
Paragraph 4: Although the topic of FMT is certainly intriguing, the authors themselves state that the virome of bacteriphages associated with it has never been investigated. I suggest to decrease the length of this chapter and limit it to the studies where the viral involvement was actually confirmed (such as the sterile filtration aspect etc.)
Response: We shortened this paragraph, as suggested.
Line 256: What is the responder/ non-responder phenotype?
Response: Responder/non-responder phenotype is now explained.
Round 2
Reviewer 2 Report
The authors have done a tremendous amount of work, which greatly improved the manuscript. In its current form it would definitely be of benefit to publish it in the journal. Only one small thing needs to be corrected:
Line 204: hepadna viruses have partial double stranded DNA genomes.